# Isolinderalactone Induces Cell Death via Mitochondrial Superoxide- and STAT3-Mediated Pathways in Human Ovarian Cancer Cells

**DOI:** 10.3390/ijms21207530

**Published:** 2020-10-13

**Authors:** Shakya Rajina, Woo Jean Kim, Jung-Hyun Shim, Kyung-Soo Chun, Sang Hoon Joo, Hwa Kyoung Shin, Seo-Yeon Lee, Joon-Seok Choi

**Affiliations:** 1College of Pharmacy, Daegu Catholic University, Gyeongbuk 38430, Korea; rajina.shakya182@gmail.com (S.R.); sjoo@cu.ac.kr (S.H.J.); 2Department of Anatomy, College of Medicine, Kosin University, Busan 49267, Korea; pangjean@naver.com; 3Department of Pharmacy, Mokpo National University, Jeonnam 58554, Korea; s1004jh@gmail.com; 4Department of Biomedicine, Health & Life Convergence Sciences, BK21 Four, College of Pharmacy, Mokpo National University, Jeonnam 58554, Korea; 5College of Pharmacy, Keimyung University, Daegu 42601, Korea; chunks@kmu.ac.kr; 6Department of Korean Medical Science, School of Korean Medicine, Pusan National University, Yangsan, Gyeongnam 50612, Korea; julie@pusan.ac.kr; 7Korean Medical Science Research Center for Healthy-Aging, Pusan National University, Yangsan, Gyeongnam 50612, Korea; 8Department of Pharmacology, Wonkwang University School of Medicine, Iksan, Jeonbuk 54538, Korea

**Keywords:** isolinderalactone, ovarian cancer, mitochondrial superoxide, janus kinase, signal transducer and activator of transcription 3

## Abstract

The mortality rate of ovarian cancer (OC) worldwide increases with age. OC is an often fatal cancer with a curative rate of only 20–30%, as symptoms often appear after disease progression. Studies have reported that isolinderalactone (ILL), a furanosesquiterpene derivative extracted from the dried root of *Lindera aggregata*, can inhibit several cancer cell lines’ growth. However, the molecular mechanisms underlying ILL activities in human OC cells remain unexplored. This study investigated the antitumor activities of ILL in human OC cells by inducing mitochondrial superoxide (mtSO) and JAK-signal transducer and activator of transcription 3 (STAT3)-dependent cell death. ILL caused cell death in SKOV-3 and OVCAR-3 cells and increased the cell proportion in the subG1 phase. Additionally, ILL significantly induced mtSO production and reduced ROS production. Moreover, ILL downregulated mitochondrial membrane potential and the expression levels of anti-apoptotic Bcl-2 family proteins and superoxide dismutase (SOD)2. Results showed that ILL decreased phosphorylation of serine 727 and tyrosine 705 of STAT3 and expression of survivin, a STAT3-regulated gene. Furthermore, ILL-induced cell death was reversed by pretreatment of Mito-TEMPO, a mitochondria-specific antioxidant. These results suggest that ILL induces cell death by upregulation of mtSO, downregulation of mitochondrial SOD2, and inactivation of the STAT3-mediated pathway.

## 1. Introduction

Ovarian cancer (OC) is a group of diseases originating in the ovaries, fallopian tubes, and peritoneum that are classified as epithelial cell carcinomas, germ cell tumors, and astrocyte stromal tumors, depending on the tissue of origin [1]. OC is the most common fatal cancer and the fifth leading cause of cancer-related death in women [2,3]. Among gynecological malignancies, the morbidity and mortality rates associated with OC are relatively high [4]. The majority of cases of early stage epithelial OC lack specific signs and symptoms, which leads to a lack of reliable screening strategies. Despite advances in surgical techniques and chemotherapy regimens, the incidence of mortality remains high for women with OC [5]. Treatment strategies against OC generally include surgical resection and subsequent platinum-based chemotherapy [6]. Although many patients initially respond well to this approach, most eventually develop recurrence of chemo-resistant disease [7]. Therefore, it is necessary to develop novel therapeutic agents for the treatment of OC.

Apoptosis, a distinct form of cell death caused by extrinsic and intrinsic stimulation, is involved in many physiological and pathological processes. Apoptotic pathways are controlled by various signaling pathways and harmonized by a network of genes [8]. Apoptosis is a crucial process to maintain cellular homeostasis and, thus, presents an important target for the development of new anti-cancer drugs [9]. Various physical and chemical stimuli that induce apoptosis evoke oxidative stress in cells [10]. During apoptosis, oxidative stress is generated both intra- and extracellularly, mainly by the generation of reactive oxygen species (ROS) [11]. Intracellular sources of ROS include mitochondrial oxidation, the microsomal cytochrome P450 system, and plasma-membrane nicotinamide adenine dinucleotide phosphate oxidases [12]. Mitochondrial dysfunction can lead to increased production of ROS, in particular superoxides, and oxygen is reduced due to uncoupling of the electron transport system, which promotes oxidative phosphorylation in cells undergoing apoptosis [13,14]. Cytotoxic superoxides induce tyrosine phosphorylation by various protein tyrosine kinases and activates various intracellular signaling pathways leading to DNA fragmentation, lipid peroxidation, and protein denaturation [15]. However, superoxide dismutase (SOD) plays an important role in the detoxification of superoxide radicals. Under physiological conditions, SOD2 (Mn-SOD) maintains an optimal level of superoxides in the mitochondria, whereas SOD1 (Cu-Zn SOD) is active in the cytosol [16]. Thus, inhibition of these SODs could lead to the accumulation of superoxides, resulting in subsequent damage to the mitochondrial membrane [17].

*Lindera aggregata*, commonly known as *Lindera*, is an aromatic evergreen shrub or small tree of the laurel family. In Asian countries, the dried roots of *L*. *aggregata* are used as a traditional medicine for the treatment of rheumatism, chest and abdominal pain, and urination problems. Moreover, *Lindera* extract is commonly used to treat the symptoms of diabetes as well as inflammation [18,19]. The main constituents of *Lindera* extract include the sesquiterpene compounds linderane, linderalactone, and isolinderalactone (ILL), which are reported to convey anti-cancer effects [20,21,22]. Among these three furanosesquiterpene derivatives, ILL was reported to exhibit greater cell death-inducing activity in human non-small lung cancer A549 cells by increasing the expression level of p21, which induces cell cycle arrest and participates in the regulation of the Fas/Fas ligand-mediated apoptosis pathway [20]. In addition, several studies have reported that ILL induced apoptosis of triple-negative breast cancer cells via downregulation of signal transducer and activator of transcription 3 (STAT3) signaling via regulation of suppressor of cytokine signaling 3 and micro-RNA 30c [21] and increased expression levels of X-linked inhibitor of apoptosis and survivin in glioma cells [23].

The aims of the present study were to assess the anti-cancer effects of ILL with the use of human OC SKOV-3 and OVCAR-3 cells and to elucidate the underlying molecular mechanisms of ILL-induced apoptosis. The study results suggest that ILL induced apoptosis in human OC cells by increasing production of mitochondrial superoxide (mtSO), decreasing the expression of SOD2, and interfering with the STAT3-mediated signaling pathway.

## 2. Results

### 2.1. ILL Induced Apoptosis in Human OC Cells

The cell viability assay was performed to assess the cell death-inducing effect of ILL. The results showed that ILL treatment inhibited the proliferation of SKOV-3 and OVCAR-3 cells in both a dose- and time-dependent manner (Figure 1A,B). ILL-induced cell death was further analyzed by Annexin V-PI staining and fluorescence-activated cell sorting (FACS) analysis. The results showed that following treatment with 20 μM ILL, 33.2% of OVCAR-3 cells were positive for Annexin V-PI-staining, while 57.6% of SKOV-3 cells were positive for PI-staining after a 48 h treatment period (Figure 1C). In addition, the phase of the cell cycle was investigated in ILL-treated cells. Moreover, ILL treatment dramatically increased the proportions of both cell lines in the subG1 phase in a dose- and time-dependent manner (Figure 1D).

### 2.2. ILL Activated the Caspase Cascade and Generated mtSO during Apoptosis

Activation of the caspase cascade is key to initiation of the apoptosis process [24]. Since ILL treatment was confirmed to induce apoptosis, activation of the caspase cascade after ILL treatment was investigated. The results showed that ILL treatment activated several members of the caspase cascade, including caspase-3, -7, -8, and -9, and also up-regulated the expression level of cleaved PARP, a well-known substrate of caspase-3 (Figure 2A,B). Therefore, we subsequently investigated whether the generation of ROS was up-regulated by ILL-induced apoptosis and found that ILL significantly decreased cellular production of ROS in a dose- and time-dependent manner, while cellular ROS was almost completely depleted in both cell lines after treatment with 20 μM ILL for 48 h (Figure 2C). Notably, ILL up-regulated the production of mtSO by more than six-fold in SKOV-3 cells and by about four-fold in OVCAR-3 cells (Figure 2D).

### 2.3. ILL Down-Regulated the Expression Levels of Proapoptotic Bcl-2 Family Proteins and SOD2

The activation mechanism of the caspase cascade is closely connected to mitochondrial proteins that control the integrity of the mitochondrial membrane [11]. Since the level of mtSO was increased by ILL, we investigated the expression levels of anti- and pro-apoptotic proteins, as well as superoxide detoxifying enzymes, in the mitochondria. The results showed that ILL treatment down-regulated the expression levels of the anti-apoptotic proteins Bcl-2 and Bcl-xL but had no effect on the expression levels of the pro-apoptotic protein Bax. Moreover, ILL reduced expression of SOD2, which is a mitochondrial SOD, while that of SOD1, a cytosolic SOD, was maintained (Figure 3A,B). Based on these results, we evaluated the mitochondrial membrane potential (MMP) as an indicator of functional mitochondrial activity after ILL treatment. As shown in Figure 3C, ILL treatment for 48 h gradually decreased MMP by 47.2% and 35.7% in SKOV-3 and OVCAR-3 cells, respectively.

### 2.4. ILL Suppressed Activation of the JAK/STAT3 Signaling Pathway

The JAK-STAT signaling pathway plays a key role in cell proliferation and apoptosis through transcriptional activation of various genes in response to cellular stimuli [25]. Hence, blocking of the JAK/STAT signaling axis could interrupt the proliferation and survival of diverse cancer cells [26]. To assess the mechanism underlying ILL-induced apoptosis, we examined the expression and phosphorylation levels of signaling molecules involved in the JAK/STAT signaling axis. The results showed that ILL decreased the expression levels of JAK2 in SKOV-3 cells in a time- and dose-dependent manner, but had no effect in OVCAR-3 cells. In dose-dependent treatment of ILL, p-JAK2 was active at lower doses (5 and 10 µM for SKOV-3 cells; 2.5 and 5 µM for OVCAR-3 cells), but not at the highest dose (Figure 4A). Next, the expression levels of p-JAK2 in response to treatment with 20 μM ILL for 12, 24, and 48 h were investigated. The results showed that p-JAK2 was activated upon treatment with ILL at 12 and 24 h, whereas the level of phosphorylation was diminished at 48 h (Figure 4B). Moreover, the expression levels and phosphorylation status of STAT3 on serine 727 and tyrosine 705 were analyzed. The results revealed that ILL reduced the levels of STAT3, p-STAT3 (Ser), and p-STAT3 (Tyr) in a time- and dose-dependent manner. In addition, ILL decreased the expression levels of survivin and STAT3 in both SKOV-3 and OVCAR-3 cells (Figure 4A,B). Next, the expression levels and phosphorylation status of protein kinases related to cell survival, including Akt, ERK, and JNK, were evaluated. Treatment with ILL decreased the expression levels of Akt in a time- and dose-dependent manner, while expression of p-Akt had increased and that of ERK remained unchanged. Moreover, expression of p-ERK increased at lower doses of ILL and decreased to the control level in response to treatment with 20 μM ILL for 48 h (Figure 4C). Furthermore, p-ERK expression increased at 12 and 24 h of treatment with ILL at 20 μM, but then returned to the control level at 48 h (Figure 4D). The protein level of JNK was stable upon treatment of ILL, while that of p-JNK was decreased in response to the highest doses of ILL (Figure 4C,D). These results suggest that ILL suppresses activation of the STAT3-mediated signaling pathway.

### 2.5. Specific Scavenging of mtSO Reverses ILL-Induced Apoptosis

Next, we investigated whether the scavenging of mtSO reversed ILL-induced cell death. Both cell lines were pre-treated with 1 or 5 nM of Mito-TEMPO, a mitochondria-targeted antioxidant, for 3 h, which was followed by treatment of SKOV-3 cells with 20 µM ILL and OVCAR-3 cells with 10 µM ILL. Pre-treatment with 5 nM Mito-TEMPO recovered the viability of SKOV-3 cells by 16.7% and 11.7%, and that of OVCAR-3 cells by 23.3% and 49.0% after ILL treatment for 24 and 48 h, respectively (Figure 5A). To confirm the ability of Mito-TEMPO to prevent cell death, activation of the caspase cascade by ILL after pre-treatment of Mito-TEMPO was investigated. As indicated by the immunoblot results, pre-treatment with Mito-TEMPO decreased the proteolytic activation of the caspase cascade (Figure 5B).

## 3. Discussion

OC is often confirmed in the advanced stage of disease due to difficulties in early diagnosis [27]. OC not only spreads into the abdominal cavity at an early stage but also has a relatively poor prognosis as compared to other cancers [28]. Therefore, the development of more efficacious treatments for OC is required. Natural products account for a large portion of current pharmaceuticals and various anti-cancer drugs, such as paclitaxel, vinblastine, and camptothecin [29]. Although linderane, linderalactone, and ILL have been identified as the active ingredients of *L. aggregata*, the anti-cancer activities and mechanisms of action have not yet been elucidated. Therefore, the aim of the present study was to assess the cell death-inducing effect of ILL in SKOV-3 and OVCAR-3 human OC cells. The results demonstrated that ILL could induce cell death through mtSO-dependent inhibition of the STAT3 signaling pathway. Many anti-cancer agents cause cell death in the form of apoptosis or necrosis. During these processes, activation of the caspase cascade executes the cell death process [30]. Here, the cell viability and flow cytometry results demonstrated that ILL initiated cell death in the form of necrosis or apoptosis depending on the cell type. In addition, ILL dramatically increased the proportions of subG1 cells (Figure 1) and activated the caspase cascade in both cell types (Figure 2A,B). Previous studies have reported that the anti-cancer activities of various natural compounds were mostly dependent on either positive- or negative-regulation of ROS production [31,32,33]. Mitochondria, which are responsible for cellular energy production, are the main subcellular organelles that produce ROS, and mitochondrial dysfunction due to ROS production is a main cause of cell death [34]. In the present study, ILL reduced the generation of total ROS in both cell types; however, the generation of mtSO was increased by about 7.5-fold in SKOV-3 cells and 4.5-fold in OVCAR-3 cells (Figure 2C,D). Moreover, the expression levels of SOD2, a mtSO-detoxifying enzyme, and the anti-apoptotic Bcl-2 family proteins Bcl-2 and Bcl-xL were reduced along with the specific increase in mtSO by ILL (Figure 3A,B). The generation of mtSOs and reduced expression of anti-apoptotic Bcl-2 family proteins by ILL resulted in the loss of MMP (Figure 3C). Several previous reports suggested that STAT3 participates in the regulation of the anti-apoptotic Bcl-2 family proteins in human OC cells [35,36]. In the present study, ILL treatment dramatically reduced the expression and phosphorylation of STAT3, as well as the expression levels of surviving, in a time- and dose-dependent manner, while ILL decreased expression of JAK2 only in SKOV-3 cells and inhibited phosphorylation of JAK2 at the highest dose (Figure 4A,B). Akt plays an important role in cell growth, survival, and death [37]. Cellular stress caused by cell death-inducing agents can dysregulate to the activities of stress-activated signaling molecules, such as the ERK and JNK pathways [38]. In this study, ILL treatment drastically reduced expression of Akt and gradually reduced that of p-JNK, but had no effect on the expression of ERK and JNK. Moreover, ILL decreased expression of p-ERK at the highest concentration in each cell type after 48 h of treatment. The increase in p-ERK at 12 and 24 h of treatment or low concentrations of ILL might be a compensating mechanism for stress induced by ILL. Selective relief of mitochondrial oxidative stress by mitochondria-specific antioxidants is known to attenuate cell death [39,40]. Here, pretreatment with Mito-TEMPO attenuated cell death in human OC cells, while ILL treatment increased production of mtSO and inhibited activation of the caspase cascade. In this study, we investigated the anticancer activity and underlying mechanism of ILL-induced cell death in ovarian cancer cells. We note that the dose of ILL used in our in vitro studies was relatively high and would be difficult to implement under physiological conditions. However, we previously showed that ILL could inhibit the growth of xenografts in a mouse xenograft model at doses of 2.5 mg/kg or 5.0 mg/kg without toxicity [23,41]. Considering the molecular weight of ILL and the dose (5 mg/kg) used in the previous animal studies to predict the achievable drug concentration of ILL at 0 time (C0), the achievable C0 is in the range of 20.5–20.47 μM assuming the volume of distribution is in the range of 1–10 L/kg. Thus, our future work will focus on lowering the effective dose of ILL using a variety of approaches, including modifying its chemical structure, to enable its use in clinical applications.

In conclusion, the results of the present study demonstrated that ILL treatment induced generation of mtSO, inactivation of the STAT3-mediated signaling pathway, and activation of the caspase cascade in human OC SKOV-3 and OVCAR-3 cells (Figure 6). These findings provide a molecular basis of ILL-induced cell death and should prove useful for the development of new ILL-based anti-OC drugs.

## 4. Materials and Methods

### 4.1. Cell Lines and Cell Culture

The human OC cell lines SKOV-3 and OVCAR-3 were purchased from the American Type Culture Collection (Rockville, MD, USA) and cultured in Roswell Park Memorial Institute (RPMI) 1640 medium (Hyclone, South Logan, UT, USA) supplemented with 10% fetal bovine serum (Hyclone) and 1% penicillin/streptomycin (Hyclone) under a humidified atmosphere of 5% CO_2_/95% air at 37 °C. All experimental procedures were carried out using 4 × 10^5^ or 7 × 10^5^ SKOV-3 or OVCAR-3 cells per 100-mm dish unless otherwise indicated.

### 4.2. Chemicals and Antibodies

ILL (cat. no. ALB-RS-6003) was purchased from ALB Technology Limited (Henderson, NV, USA), prepared as a 100 µM stock solution in dimethyl sulfoxide (Sigma Chemical Company, St. Louis, MO, USA), aliquoted, and stored at −80 °C before use. Primary antibodies against all cleaved caspases, cleaved poly (adenosine diphosphate-ribose) polymerase (PARP), Bax, SOD1, Akt, mitogen-activated protein kinase (ERK), C-Jun N-terminal kinase (JNK), phosphorylated (p)-AKT, p-ERK, p-JNK, STAT3, p-STAT3 (Ser), and p-STAT3 (Tyr) were purchased from Cell Signaling Technology (Danvers, MA, USA), while antibodies against Bcl-2, Bcl-xL, and SOD2 were obtained from Santa Cruz Biotechnology, Inc. (Santa Cruz, CA, USA), and antibodies against survivin were acquired from Novus Biologicals LLC (Centennial, CO, USA). All secondary antibodies were obtained from Thermo Fisher Scientific (Waltham, MA, USA).

### 4.3. Cell Viability Assay

The viability of SKOV-3 and OVCAR-3 cells was assayed using the QuantiMax^TM^ WST kit (BioMax Co., Ltd., Seoul, Korea). Briefly, 10^4^ cells were seeded into quadruplicate wells of a 48-well plate and incubated for 24 h at 37 °C under an atmosphere of 5% CO_2_/95% air. Afterward, the cells were treated with 0, 5, 10, 20, or 50 µM of ILL for 24 or 48 h. Water soluble tetrazolium (WST) cell proliferation reagent was prepared as five-fold dilutions in RPMI media. Following ILL treatment, 200 µL of WST solution were added to each well and the plate was incubated for an additional 2 h. Afterward, the optical density of the wells was measured at an absorbance of 450 nm using a FLUOstar^TM^ Omega microplate reader (BMG Labtech, Ortenberg, Germany).

### 4.4. Measurement of Cell Death and Assessment of the Cell Cycle Phase after ILL Treatment

For dose-dependent cell death analysis, SKOV-3 cells were treated with 0, 5, 10, and 20 µM ILL for 48 h, while OVCAR-3 cells were treated with 0, 2.5, 5, and 10 µM ILL for 48 h. For time-dependent cell death analysis, SKOV-3 cells were treated with 20 µM ILL for 12, 24, and 48 h, while OVCAR-3 cells were treated with 10 µM ILL for 12, 24, and 48 h. Post-treatment, the cells were harvested by trypsinization, washed twice with filtered phosphate-buffered saline (PBS), stained with the use of a fluorescein isothiocyanate-Annexin V apoptosis detection kit (BD Biosciences Pharmingen, San Diego, CA, USA), and subjected to flow cytometry with the use of a FACSCalibur flow cytometer (BD Biosciences, San Jose, CA, USA). The results were analyzed using CellQuest Pro™ software (Becton, Dickinson and Company, Franklin Lakes, NJ, USA). For cell cycle analysis, both cells types were treated with ILL in the same way as for cell death analysis. Following ILL treatment, the cells were trypsinized, washed twice with filtered PBS, and fixed overnight in 5 mL of ice-cold 70% ethanol at 4 °C. The next day, the cells were washed three times with filtered PBS and labeled with 50 μg/mL of propidium iodide (PI) (Life Technologies, Carlsbad, CA, USA) and 100 μg/mL of RNAase A (Biosesang, Daejeon, Korea) in filtered PBS. After a 30-min incubation period at 37 °C, the proportion of subG1 cells was determined by flow cytometry using a FACSCalibur flow cytometer.

### 4.5. Measurements of ROS and mtSO Contents, and MMP

SKOV-3 cells were treated with 0, 5, 10, and 20 µM ILL for 12, 24 and 48 h, while OVCAR-3 cells were treated with 20 µM ILL for 12, 24, and 48 h. Afterward, the cells were trypsinized, washed twice with filtered 0.1% bovine serum albumin in PBS, and stained at room temperature with specific indicators for measuring different parameters. To determine the cellular ROS content, the cells were stained with 10 µM 2′ 7′-dichlorodihydrofluorescein diacetate (DCF-DA) (Life Technologies) for 1 h. To determine the mtSO content, the cells were stained with 5 µM MitoSOX Red reagent (Thermo Fisher Scientific) for 30 min. For MMP assessment, the cells were incubated with 3,3′-dihexyloxacarbocyanine iodide (Life Technologies) for 30 min and then subjected to flow cytometry using a FACSCalibur flow cytometer. The data were analyzed using CellQuest Pro™ software.

### 4.6. Immunoblot Analysis

After the indicated ILL treatments, the cells were harvested, washed with PBS, and lysed with RIPA buffer containing 1 mM sodium orthovanadate, 1 mM NaF, 0.1 mM phenylmethylsulfonyl fluoride, and a protease inhibitor cocktail. The protein yield was measured using the Pierce^TM^ Coomassie (Bradford) Protein Assay Kit (Thermo Fisher Scientific) in accordance with the manufacturer’s instructions. Briefly, equal amounts of protein were aliquoted, denatured for 5 min by boiling, loaded into the wells of 10–15% gels, and separated by sodium dodecyl sulfate-polyacrylamide gel electrophoresis. Afterward, the proteins were transferred onto a polyvinylidene difluoride membrane (EMD Millipore Corporation, Billerica, MA, USA), which was blocked with 5% skim milk in Tris-buffered saline–Tween^®^ 20 detergent (TBST) for 1 h at room temperature. Then, the membrane was washed briefly with TBST and incubated with diluted primary antibodies (diluted to 1:1000 in 2% skim milk in TBST) overnight at 4 °C. The next day, the membrane was incubated with the secondary antibodies (diluted to 1:5000 in 2% skim milk in TBST) for 2 h at room temperature. The membrane-bound proteins were visualized using enhanced chemiluminescence reagent (GE Healthcare Life Sciences, Chicago, IL, USA).

### 4.7. Statistical Analysis

Data are presented as the mean ± standard deviation. Single comparisons were performed using the Student’s *t*-test. Data analysis was performed using SigmaStat 3.5. statistical software (Systat Software, Inc., San Jose, CA, USA). A probability (*p*) value of <0.05 was considered statistically significant.

## Figures and Tables

**Figure 1 ijms-21-07530-f001:**
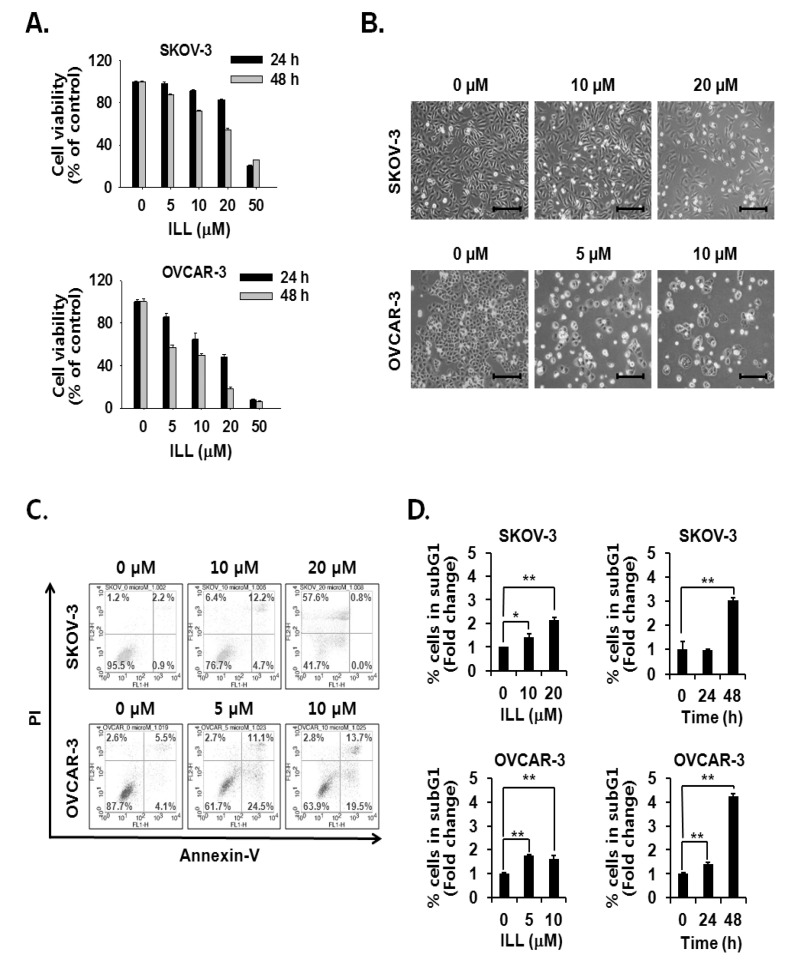
Cell death-inducing effects of isolinderalactone (ILL) in human ovarian cancer (OC) cells. (**A**) ILL inhibited cell viability in a time- and dose-dependent manner. Cell viability was assessed with the water-soluble tetrazolium (WST) assay. (**B**) Morphological changes to SKOV-3 and OVCAR-3 cells in response to ILL treatment after 48 h. Microscopic images after 48 h. Scale bar = 200 μm. Magnification, 100×. (**C**) ILL increased the proportion of PI-positive SKOV-3 cells and Annexin V-positive OVCAR-3 cells. The data are presented as dot plots. (**D**) ILL enhanced the proportion of subG1 cells in both cell types. Data are presented as the mean ± SD. * *p* < 0.05 and ** *p* < 0.001 vs. the control group.

**Figure 2 ijms-21-07530-f002:**
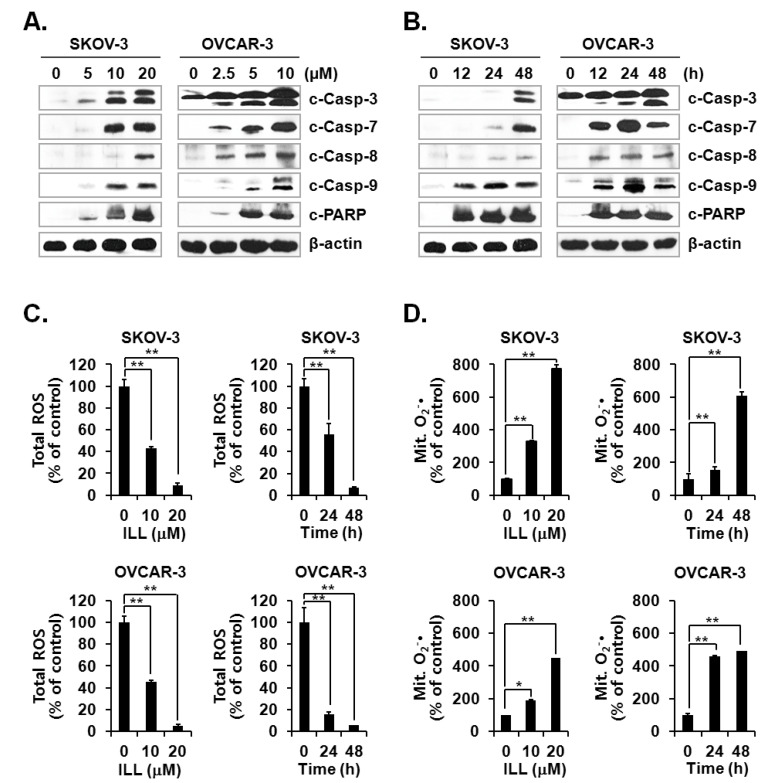
Activation of caspases and changes in cellular and mitochondrial ROS levels after ILL treatment. (**A**,**B**) ILL treatment proteolytically activated caspase-3, -7, -8, and -9, and cleaved poly (adenosine diphosphate-ribose) polymerase (PARP) in a time- and dose-dependent manner. Intracellular levels of reactive oxygen species (ROS). (**C**) and mitochondrial superoxide (mtSO). (**D**) were detected by dichlorodihydrofluorescein diacetate (DCF-DA) and mitoSOX using fluorescence-activated cell sorting (FACS). Values are presented as the mean ± SD of triplicate samples. * *p* < 0.05 and ** *p* < 0.001 vs. the control group.

**Figure 3 ijms-21-07530-f003:**
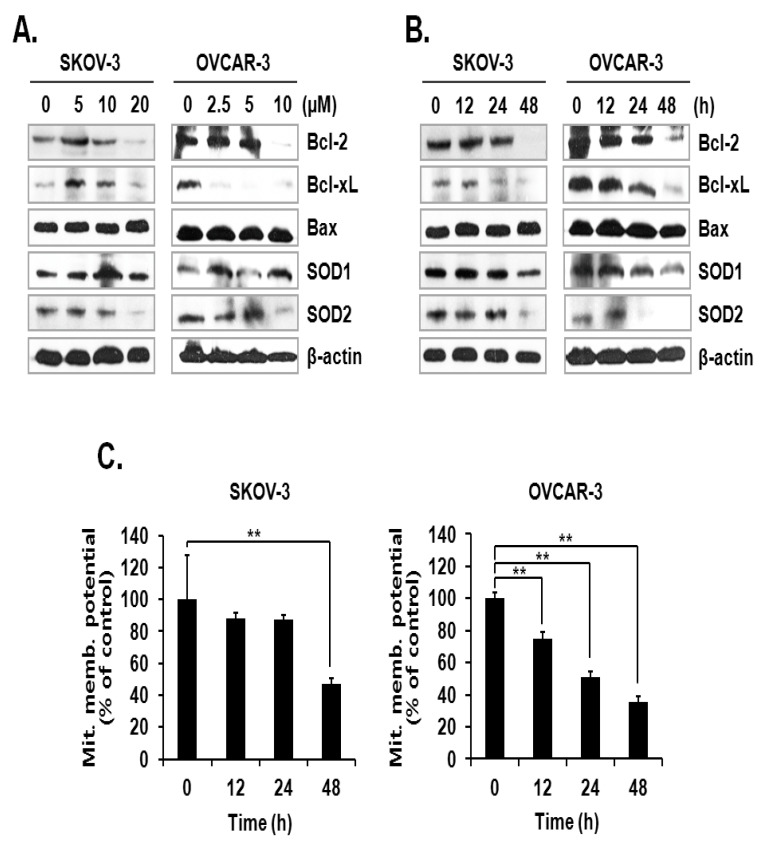
Loss of anti-apoptotic Bcl-2 family proteins, SOD2, and MMP after ILL treatment. (**A**,**B**) ILL decreased cellular levels of Bcl-2, Bcl-xL, and SOD2 (immunoblot analysis). (**C**) Changes in MMP after ILL treatment detected by flow cytometry. ILL diminished MMP in SKOV-3 and OVCAR-3 cells in a time- and dose-dependent manner. Values are presented as the means ± SD of triplicate samples. * *p* < 0.05 and ** *p* < 0.001 vs. the control group.

**Figure 4 ijms-21-07530-f004:**
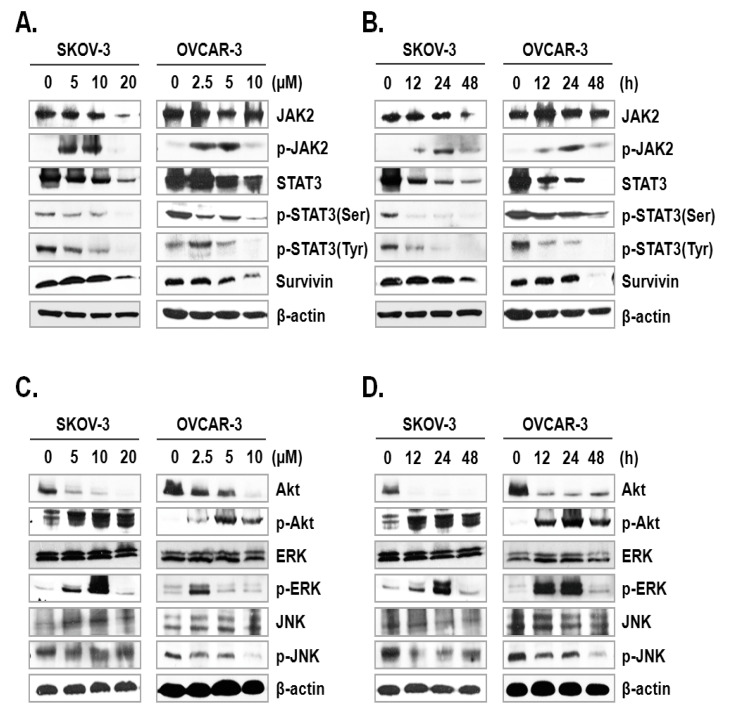
ILL-induced inhibition of STAT3-mediated signaling and gene expression. (**A**,**B**) SKOV-3 and OVCAR-3 cells were treated with the indicated ILL concentrations for the indicated times. Cellular levels of p-STAT3 (Ser), p-STAT3 (Tyr), and STAT3 were determined by immunoblot analysis. (**C**,**D**) Expression levels and phosphorylation status of various stress-activated kinases detected by immunoblot analysis.

**Figure 5 ijms-21-07530-f005:**
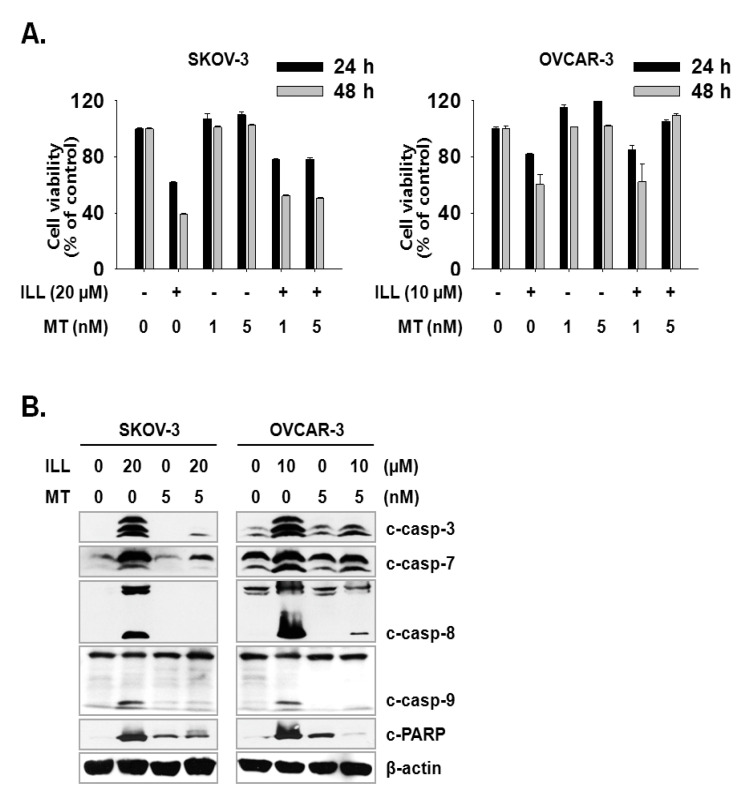
Amelioration of cell death by ILL using mtSO-specific scavenger. (**A**) The pretreatment of Mito-TEMPO rescued ILL-induced cell death. Both cells were pretreated with 1 or 5 nM of Mito-TEMPO for 3 h before 24 h or 48 h of ILL treatment. (**B**) Mito-TEMPO prevented proteolytic activation of caspases and cleavage of PARP. The levels of each protein were detected by immunoblot. Values are the means ± SD for data in triplicate.

**Figure 6 ijms-21-07530-f006:**
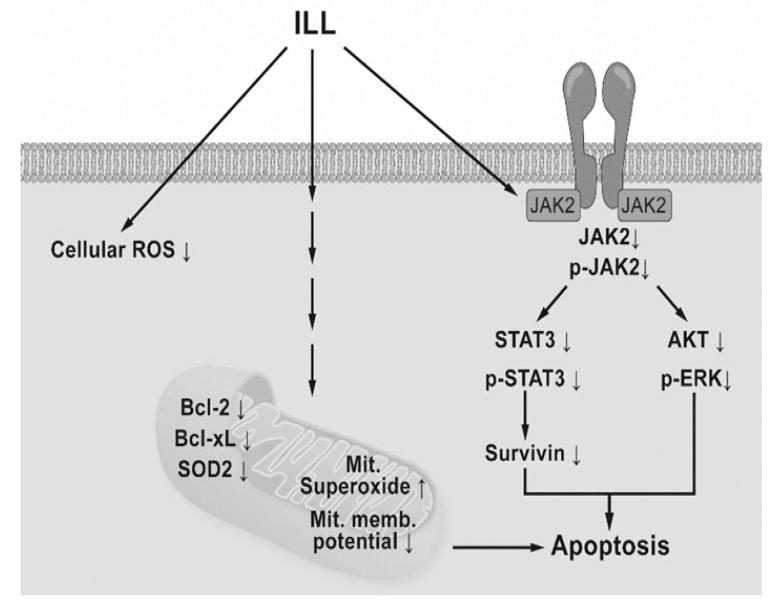
Schematic diagram of ILL-induced cell death. ILL induces generation of mtSO and decreases MMP in both cells through reduction of Bcl-2, Bcl-xL, and SOD2 expression. On the other hand, ILL declines in expression and phosphorylation of STAT3 and survivin, a STAT3-regulated gene during cell death.

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
