# Peer review of "Isolinderalactone Induces Cell Death via Mitochondrial Superoxide- and STAT3-Mediated Pathways in Human Ovarian Cancer Cells"

_ijms, 2020, doi:10.3390/ijms21207530_

Round 1

Reviewer 1 Report

In the article, authors used isolinderalactone from Lindera aggregata to demonstrates its anti-cancer properties. However, the author used concentrations in 5, 10, 20, and 50 uM to demonstrate the mechanism of action. However, these concentrations are supraphysiological concentrations. Physiologically it is very difficult to attain uM concentrations of drugs. Drugs may reach maximum concentration in uM but average concentrations are in the nM range. Therefore, the authors need to justify the rationale for choosing such high concentrations. If physiological concentrations can not reach these level then the anticancer mechanism described by authors are not possible in humans. The authors need to demonstrate mechanistic pathways in physiologically relevant concentrations. 

In certain cases, drugs do reach high concentrations, if authors believe that this drug can reach high such concentrations at target organs (ovaries). Then authors need to provide the rationale based on the physicochemical properties of drugs and experiments in the animal models to demonstrate these effects.

Authors also need to include positive and negative controls in the experiments. 

Reviewer 2 Report

The authors reported the mechanistic effects of ILL, a derivative from a plant, in ovarian cancer cell lines. They found ILL induced mitochondrial ROS and suppressed multiple oncogenic pathways especially Jak2/STAT3. The methodology and design are scientifically sound. Minor comments are as follows.

  1. The concentration of ILL is in 10-20 uM, which may not be achievable in real human body. The authors may discuss this limitations and possible solutions.

Round 2

Reviewer 1 Report

Concentrations used by the author are significantly higher than physiological concentrations. The authors discussed concentrations by justifying that in previous studies ILL induced cell death at 2.5mg/kg or 5mg/kg dose and 10 mg/kg was well tolerated in mouse to 27 days. However, authors need to justify concentrations used for in-vitro data based on PK data from mouse models. Without knowing PK data, we can not assume how much will be bioavailability and plasma concentrations. 

Author Response

We thank again the reviewers for their progressive advices.

The purpose of this study is to demonstrate the anti-cancer effect of ILL against human ovarian cancer cells. In this study, the dose of ILL we used was obtained from the results of time- and dose-dependent cell viability test for both human ovarian cancer cells. The dose of ILL used in previous animal study of human glioblastoma was not determined through PK modeling, but was established within the dose range which ILL toxicity does not appear.

In general, the volume of human body fluid is 0.6 L/kg (40 L/70 kg). The molecular weight of ILL was 244.28, and the maximum dose of ILL used in the previous mouse xenograft model of glioblastoma was 5 mg/kg. Assuming that the volume of distribution for ILL in mice is from 1 to 10 L/kg, the drug concentration of ILL at 0 time (C0) could be 2.05-20.47 mM, and the dose of ILL used for this study was included within this range.

Therefore, the dose of ILL used in this study is relatively high compared to other compounds with anticancer activity, however, it is not a dose that cannot be achieved clinically. In addition, we are trying to increase the anticancer activity and decrease the dose of ILL through chemical modification, and also plans to promote joint research with Dr. Beom Soo Shin who is studying about PK/PD modeling of natural compounds (Molecules 23(2):349, Molecules 22(9):1488).

In addition, we revised the discussion part of the manuscript to reflect the editor's opinion, and the revised contents are marked in red.
